# Cognition in Children with Arachnoid Cysts

**DOI:** 10.3390/jcm9030850

**Published:** 2020-03-20

**Authors:** Ulrika Sandvik, Tomas Adolfsson, Dan N. Jacobson, Kristina Tedroff

**Affiliations:** 1Section for Neurosurgery, Department of Clinical Neuroscience, Karolinska Institutet, 171 77 Stockholm, Sweden; 2Neuropediatric Unit, Astrid Lindgren Children’s Hospital, Karolinska University Hospital, 171 76 Stockholm, Sweden; tomas.adolfsson@sll.se; 3Department of Women’s and Children’s Health, Karolinska Institutet, 171 77 Stockholm, Sweden; dan.jacobson@ki.se (D.N.J.); kristina.tedroff@ki.se (K.T.)

**Keywords:** arachnoid cyst, cognition, neuropsychology, pediatric neurosurgery, neurosurgery

## Abstract

Background: This study aims to evaluate if children with temporal arachnoid cysts (AC) have cognitive symptoms and if neurosurgery improves these. Methods: A prospective case series study including consecutive pediatric patients with temporal AC. The children underwent neuroradiology, neuroopthalmologic evaluation, and a standard electroencephalography (EEG). Additionally, a neuropsychologist performed a standardized set of evaluations, with a one-year follow-up consisting of Weschler Intelligence Scale for Children version IV (WISC-IV), FAS (for verbal fluency), Boston Naming Test (BNT, for visual naming ability) and NEPSY-II (Developmental NEuroPSYchological Assessment) for verbal memory. Results: Fifteen children, 9 boys and 6 girls, were evaluated and 11 underwent surgery. The Full Scale IQ subscore (FSIQ) improved from M = 84.8 to M = 93.0 (*p* = 0.005). The preoperative Verbal Comprehension Index (VCI) was in the low average range (M = 86.7), improving to a level within the average range (M = 94.7, *p* = 0.001). Preoperative Perceptual Speed Index (PSI) was in the below average range (M = 81.5), improving to a level within the average range (M = 92.5, *p* = 0.004). Conclusion: ACs are a common finding in a pediatric neurosurgical setting. Our data suggest that some temporal AC have a negative effect on general cognitive ability and that this impairment can be improved by surgery. We suggest a standardized evaluation, including comprehensive and validated neuropsychological assessment tools, to thoroughly assess symptoms as well as the postoperative outcome.

## 1. Introduction

Arachnoid cysts (ACs) are benign, space-occupying, intracranial, or intraspinal anomalies that are most commonly identified in childhood. They are typically considered congenital lesions [1] but a small proportion is believed to be secondary to trauma, inflammation, or bleeding [2]. 

Arachnoid cysts are common, with a reported prevalence of 2.6% when ACs have been identified in children who have undergone neuroradiological examinations for reasons other than suspected ACs [3]. The majority of individuals where ACs have been identified are male and the middle fossa is the most common location [3]. Temporal ACs have been linked to delayed psychomotor development, seizures, headache, hydrocephalus, and cranial deformation [4]. Additionally, in case series, temporal cysts have been associated with dizziness, decreased visuospatial orientation, imbalance, as well as impairment of higher cognitive functions such as memory [4,5,6,7].

Whether or not ACs in children with possible symptoms should be operated has been a matter of long-standing controversy. Some studies claim that ACs are to be regarded as congenital findings that should not be surgically treated unless they cause severe symptoms [5]. While others suggest a connection between temporal ACs and neuropsychiatric symptoms motivating surgical treatment to possibly alleviate these symptoms [4].

However, the limited amount of published data, the heterogeneity of the available studies, as well as the shortage of information retrieved from standardized and validated cognitive and psychological pediatric assessment tools, merits further investigation. In our clinical experience, parents of children who have undergone surgery for temporal ACs often express unexpected cognitive gains of surgery such as increased ability to concentrate and improved school results. Thus, we designed a prospective consecutive case series study aiming to describe and evaluate if children with temporal ACs had neuropsychiatric or cognitive signs or symptoms possibly attributed to the ACs and if these symptoms changed when surgery was performed.

## 2. Materials and Methods

This is a prospective case series including consecutive children aged 6–13 years with temporal AC evaluated at Astrid Lindgren Children’s Hospital, Karolinska University Hospital, a tertiary hospital in Stockholm, Sweden (patient characteristics Table 1). All children were referred to the pediatric neurosurgery outpatient clinic due to neurological, neuropsychiatric, or endocrine symptoms possibly attributed to a known temporal AC. The cyst had been identified through a standard investigation with magnetic resonance imaging (MRI) or computer tomography (CT). Patients were evaluated by a pediatric neurosurgeon and a pediatric neurologist. If symptoms and neuroradiology findings merited further investigation, a standard workup of, a three Tesla MRI, a neuro-ophthalmologic evaluation also including visual fields, and a standard surface electroencephalography (EEG) was performed.

Specifically, for this study, children were evaluated by a pediatric neuropsychologist who performed a standardized set of validated, age-adjusted tests using the same test versions consistently throughout the project. The following tests were included.

### 2.1. Weschler Intelligence Scale for Children 

Weschler Intelligence Scale for Children version IV (WISC-IV) is an individually administered instrument used to assess intellectual functioning in children 6–16 years of age. WISC-IV consists of 10 core subtests that contribute to the full-scale IQ (FSIQ). The ten core subtests are also used to derive four indexes measuring Verbal Comprehension Index (VCI), Perceptual Reasoning Index (PRI), Working Memory Index (WMI), and Processing Speed Index (PSI) [8]. 

### 2.2. FAS

Verbal abilities were assessed using the FAS, a subtest of the Neurosensory Center Comprehensive Examination for Aphasia [9]. FAS is a test of phonological verbal fluency and requires the test subject to produce as many words as possible beginning with the letters F, A, and S in 60 s each. 

### 2.3. Boston Naming Test

The Boston Naming Test (BNT) is a measure of visual naming ability. The test subject is required to name 60 objects from black and white drawings [10].

### 2.4. NEPSY-II

The NEPSY-II (A Developmental NEuroPSYchological Assessment) [11], is used to evaluate the child’s neuropsychological development in five functional domains. We used NEPSY-II in this study to assess verbal memory as measured by retention of a story read by the examiner and memory for a list of 15 words read by the examiner, with retention immediately after the presentation and after a 30-min delay. Visual memory was tested using memory for 16 faces, with retention immediately after the presentation and after a 30 min delay.

### 2.5. Rey Complex Figure Test

Rey Complex Figure Test (RCFT) is a test of visuospatial and visuoconstructive abilities. It also gives a good indication about executive function [12].

### 2.6. Nordic Five-to-Fifteen

The parent’s perception of their child’s daily life functioning—that is, information about the child’s strengths and weaknesses, as well as developmental level relative to that of peers, were evaluated through the Nordic Five-to-Fifteen-questionnaire (FTF) [13]. This assessment tool was not included at the study start but added to the protocol later, with the aim to evaluate if FTF could be a beneficial instrument in future studies. The FTF consists of 181 statements arranged into eight different domains (memory, learning, language, executive functions, motor skills, perception, social skills, and emotional/behavioral problems), most of which can be subdivided into subdomains. The FTF is often used in the Nordic countries and normative age and gender data are available. The FTF is available in eight languages including English [13].

### 2.7. Additional Evaluations

Neuroradiological imaging was performed with a 3 Tesla magnetic resonance imaging (MRI) before and six months after the surgery. The volume of the cysts pre- and postoperatively were calculated using neuronavigational software (Iplannet, Brainlab, Germany) on axial T2 weighted MRI-images by one investigator. The cysts were further divided into type I-III according to the Galassi classification [14].

### 2.8. Clinical Decision Process

When neuropsychological or cognitive findings corresponding to cyst location were identified, surgery was proposed. Additional factors included in favor of surgery were severe headache and epilepsy with EEG verified ipsilateral focus.

The surgical method used in this study is an open micro-neurosurgical fenestration. This is the method of choice at our department due to its efficacy and low complication rate [15]. The procedure was performed through a standard temporal craniotomy. The thickened arachnoidea that forms the cyst walls was resected superficially and the deeply located arachnoid layers around the internal carotid artery and oculomotor nerve incised to allow cerebrospinal fluid circulation (Figure 1).

Postoperatively, within 10–12 months’ time-frame after surgery, the following evaluations were repeated; a three Tesla MRI brain scan, clinical neurological evaluation, and the neuropsychology assessments. Evaluation by an ophthalmologist and EEG were only repeated if preoperative assessments had been abnormal.

All parents and children were given oral and written study information and consented to participation and the publication of the results. The study was approved by the Regional Ethics Board in Stockholm (EPN 2015/268-31/1) and adhered to the Helsinki declaration.

### 2.9. Statistical Analysis

Changes in pre- and postoperative cyst volumes as well as the comparison of cyst volume size and WISC-IV were tested using a paired t-test.

The pre- and postoperative WISC-IV results (FSIQ, VCI, PRI, WMI, and PSI) as well as NEPSY-II were analyzed using paired t-tests. Pre- and postoperative FAS, BNT and RCFT results were analyzed using the Wilcoxon signed-rank test due to non-normally distributed data. FAS and BNT were, additionally, standardized to age and pre- and postoperative differences analyzed using a paired t-test. Both analyses of FAS and BNT are presented. Possible interactions between test re-test time interval and change in FSIQ were explored using Pearson’s correlation coefficient (PCC).

Statistical significance level was pre-set at p < 0.05 The data was analyzed using SPSS version 22 and Stata 14 IC.

## 3. Results

Fifteen children, six girls and nine boys, 6–13 years old, mean age 9.5 (SD ± 2.0) years were consecutively referred and evaluated for inclusion. Table 1 presents patient characteristics.

Two children (children 14 and 15) displayed only mild symptoms and had normal neuropsychological assessments. Hence, they were not offered surgery. Thirteen of the 15 children were recommended surgery. All families, but one, accepted the intervention. For the twelve children that underwent elective surgery, the neuropsychological evaluation displayed findings concordant with the cyst location and cyst laterality. One of the operated children (child 12, Table 1) had a preoperative FSIQ well below 70. This child was excluded from the analysis since establishing IQ correctly is more difficult and uncertain when in the lower range [16,17]. Hence, the cognitive data for this child is not presented. The conservatively managed children did not undergo radiological nor neuropsychological follow-up.

The mean preoperative cyst volume was 68.1 mL (range 13–341 mL SD 92.6 mL) and the mean postoperative volume was 37.3 mL (range 3–207 mL, SD 55.4 mL). The cyst size was reduced by an average of 45% (*p* = 0.01). Due to discomfort and claustrophobia in the MR setting, one patient underwent CT as a postoperative scan.

During the hospital stay, and later in the home setting, no postoperative complications were noted. In one situation, where the child had a rather large cyst (341 mL), the fenestration proved insufficient, and a second procedure was performed with the insertion of a cystoperitoneal shunt. For this child (child 6, Table 1), postoperative cyst volume was calculated from the post shunting images.

Neuropsychological tests were re-administered on average 11.6 months (SD 3.2) after the surgical intervention (Table 2).

### 3.1. WISC-IV

When evaluated with WISC-IV, our sample performed preoperatively in the low average range with FSIQ (M = 84.8, SD 8.7). (Table 2) Improvement in FSIQ was noted postoperatively, with results within the average range (M = 93.0, SD 11.9), corresponding to a mean score improvement of 8.2 (95% CI 3.2–13.2, *p* = 0.005). The preoperative VCI was in the low average range (M = 86.7, SD 13.1), improving postoperatively to a level within the average range (M = 94.7, SD 13.1) (*p* = 0.001). The difference between baseline PRI (M = 99.1, SD 11,4) and postoperative PRI (M = 103.8, SD 9.6) did not reach statistical significance. Preoperative PSI was below average range (M = 81.5, SD 10.4), improving postoperatively, to a level within the average range (M = 92.5, SD 12.7, *p* = 0.004). The pre- and postoperative WMI were both below average range and showed no improvement (M = 82.5, SD = 9.65, and M = 84.5, SD = 15.70). There was no statistically significant interaction between test—re-test time interval and WISC-IV results (PCC −0.51). No significant correlation between the postoperative improvement measured by FSIQ and reduction in cyst volume could be seen (*p* = 0.06). Significant findings were, however, seen between decreased cyst volume and improvements in PSI (*p* < 0.001) and VCI (*p* < 0.001).

### 3.2. NEPSY-II

In the memory assessment with NEPSY-II, the results for visual memory (NEPSY-II Memory for faces) improved postoperatively for both immediate (*p* = 0.033) and delayed (*p* = 0.016) recall. (Table 2) There was also an improvement in the results for verbal memory (memory for a list of words) in the mean number of words recalled between preoperative (M = 10.71) and postoperative assessment (M = 12.7) (*p* = 0.027). The results for NEPSY-II (memory for a story read by the examiner) were unchanged.

### 3.3. FAS

On the test of verbal fluency (FAS), the number of words generated did not improve after surgery, neither in raw scores or age-standardized scores.

### 3.4. BNT

The preoperative raw scores on picture naming (BNT) (Median= 35, Range 22–44) were improved postoperatively (Median = 33, Range 31–45) (*p* = 0.023), mainly due to significant improvements in the participants with the lowest performance pre-operatively, however, no significant improvement was seen on age-standardized BNT-scores (Table 2).

### 3.5. RCFT

When visuoconstructive ability and visuospatial memory (RCFT) was tested, no postoperative changes were seen neither in the copy condition nor in the memory condition, with retention after a three-minute delay and after a 30 min delay (Table 2).

### 3.6. Opthalmology and EEG

All children were evaluated by an ophthalmologist. None of them had visual field defects, papillary edema or pathological findings when retina was evaluated by optical coherence tomography. Two of the children had verified epilepsy and three displayed pathological EEG-findings.

## 4. Discussion

In our study, a marked improvement was seen compared to baseline in FSIQ, for the children that had neurosurgery forthe AC. We also identified improvements in verbal function and processing speed, abilities attributed to the temporal lobe, and important factors for educational as well as social functioning. The cognitive improvements were seen on tests that specifically evaluated verbal functions. Tests evaluating frontal and parietal functions did not show similar improvements.

Seven out of eleven children that ultimately had surgery, and were included in the analysis, first presented with symptoms of learning, concentration, perception, or language difficulties.

Throughout the last decades, there have been several papers reporting possible associations between temporal ACs and cognitive symptoms [5,6,7,18,19]. Only a few studies have systematically investigated cognition in larger groups of patients with cysts before and after surgical decompression, and most of these studies have been conducted in adults. The studies that do report different aspects of cognitive functioning before and after neurosurgical intervention do seem to confirm that ACs may impair cognition and that the preoperative cognitive symptoms might be improved or even normalized following surgical decompression [6,18]. Visuospatial defects, impairment in executive function and memory have shown to be improved by surgery of temporal cysts in several case series [4,6,7].

As IQ is a common global measure of cognitive functioning, it has been constructed so that IQ 100 corresponds to the average cognitive level (95% CI 70–130) and an IQ of <70 corresponds to intellectual disability. Much simplified FSIQ levels of 85 or more are essential to comprehend and learn in a standard school situation. Thus, the improvement of close to 8 points in FSIQ in this study is likely to be of clinical importance for the individual child. A recent study in 24 children, with a mean age of 9.4 years, with temporal, frontal, and frontotemporal cysts did not show improved intellectual abilities after surgery [20]. However, the younger children, under 7 years of age, did display an improvement in FSIQ, verbal comprehension testing, an indication of the possible greater recovery rate in the still more developing brain [20]. In our study, the WISC-IV test was repeated after an average interval of 11.6 months after the initial testing. This is somewhat shorter than the often recommended one year to exclude a practice effect. A recent study, with eight-year-old American children tested with WISC-IV 10 months apart, indicated that the FSIQ is the most stable score yielded by the WISC-IV, with a stability coefficient of 0.88 or excellent [21]. No significant interaction between test—re-test time interval and FSIQ improvement was seen in our study. The other tests have not, to the best of our knowledge, been described to have any significant practice effects over time.

Moreover, improvements were seen in verbal functions and processing speed. Processing speed is a cognitive factor important for attention and how effectively and rapidly other cognitive resources can be used in different cognitive tasks [22]. The processing speed has great importance for school work as well as social interactions [23]. Gordon and coworkers conclude that processing speed is central for other executive functions, e.g., working memory and inhibition [24].

Other cognitive functions such as perceptual functions and working memory functions were not as affected preoperatively, possibly explaining why significant improvements were absent. This can be explained by the temporal location of the cysts. Working memory is considered to mostly be a frontoparietal lobe function while verbal fluency, here measured by FAS, is primarily considered to be a frontal brain function. The parental questionnaire 5–15, was introduced late in the study and only six of the parents completed the FTF. Consequently, no statistical analyses were performed. We believe that FTF is a useful tool appreciated by parents that can be recommended for future studies in this area.

Cyst size decreased on average 45% (*p* = 0.01) after surgery, however, no correlation between a reduction in cyst size and improvement in FSIQ could be seen. This is in accordance with study results hypothesizing that it is the intramural cyst pressure, rather than the cyst size that might cause symptoms [25]. Accordingly, several other studies have failed to demonstrate a correlation between clinical improvement and a reduction in cyst volume [26,27,28,29]. However, in our material, we could see an apparent correlation between improved VCI and PSI subscores and reduction in cyst volume. In a Norwegian report of 48 children, a significant association was seen between cyst volume reduction and clinical improvement when symptoms of headache and epilepsy, as well as self-reported level of function, were assessed [30]. Apparently, the correlation of symptoms attributed to the cyst as well as the effect of cyst reduction is not well understood

Before surgery, two of the children were suffering from epilepsy. Both had pathological EEG-findings concordant to the cyst location and both were seizure-free at follow up and up to two years after the surgical procedure. Epilepsy is often associated with AC but the seizure etiology remains unknown and the connection between cysts and seizures has sometimes been suggested to be incidental [29,31,32]. Some authors state a temporal predilection for cyst related to epilepsy and describe seizure frequencies as high as 31% in patients with temporal cysts [33]. A large pediatric study (*n* = 81) showed epilepsy in 22% of children with ACs (*n* = 81) [34]. The same study displayed EEG abnormalities in 81% of the children where an EEG had been performed [34].

In our material, microneurosurgical fenestration seems to be a safe and effective procedure. Out of the eleven children that had surgery, one child only needed re-operation with a cystoperitoneal shunt due to insufficient fenestration effect. Acute postoperative expected symptoms included swelling, difficulties in chewing, headache and nausea were present in all children. These effects were transient ceasing after 1–2 weeks, and there was no report of late-onset complications. Children typically stayed in the hospital for four days after surgery, all returning to school within two weeks of surgery.

Based on these findings, we propose that children with temporal AC and a history of epilepsy, headache, or cognitive difficulties should be evaluated by a neuropsychologist familiar with standardized clinical tools assessing temporal lobe functions. Our results are supported by a recent prospective study in adults [35]. A cognitive profile suggesting impaired temporal lobe functions corresponding with the laterality of the cyst should be considered as an indication for surgery and the pros and cons with microneurosurgical fenestration should be discussed with the family.

This study is, to the best of our knowledge, the first study where children with AC have been prospective and systematically evaluated with a wide range of standardized and validated cognitive tools before and after a neurosurgical procedure.

### Limitations of the Study

The small sample size and a design lacking prospective follow-up of controls are limiting factors and make testing of existing working theories difficult [36,37]. However, any study design that includes a control population that has either declined surgery or had surgery performed at a later time will be at risk of introducing selection bias. Another possible shortcoming is that external validity can be low if findings are applied to cysts with other intracranial locations than the temporal lobe [38,39,40,41]. The sample size is also too small to allow any conclusions regarding the laterality of the cyst.

## 5. Conclusions

Arachnoid cysts are a rather common finding in the pediatric neurosurgical setting. Our data suggest that some temporal AC might have a negative effect on general cognitive ability and that this impairment could be improved by microneurosurgical fenestration. We suggest a standardized set of evaluations conducted by the neurosurgeon, pediatric neurologist, and neuropsychologist to thoroughly assess any symptoms as well as the postoperative outcome.

## Figures and Tables

**Figure 1 jcm-09-00850-f001:**
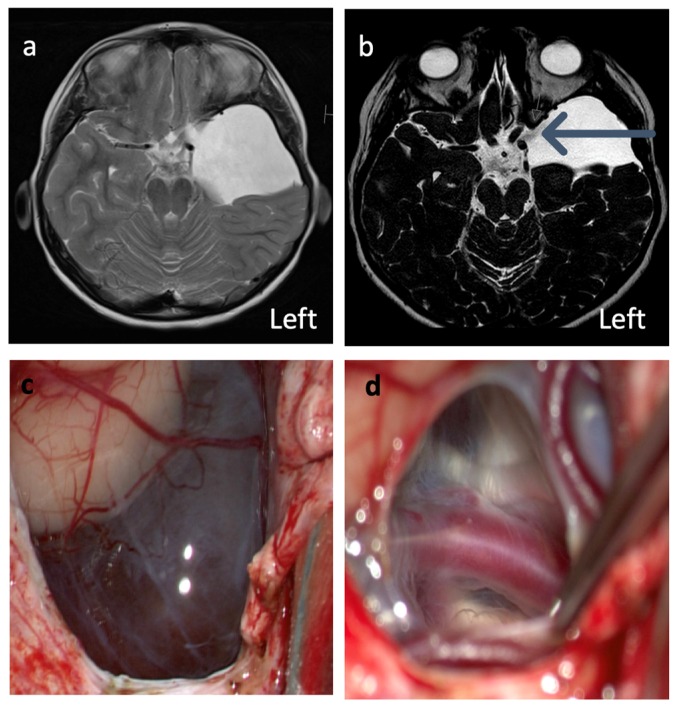
MRI images of left sided temporal AC (patient 1, Table 1) before (**a**) and after (**b**) fenestration. In image b, the fenestration is seen as a flow void (arrow). Image (**c**) displays a cyst after the opening of dura and image (**d**) shows the inner cyst lining above the medial cerebral artery and the cranial nerves.

**Table 1 jcm-09-00850-t001:** Patient characteristics.

Patient ID	Sex	Presenting Symptoms	Cyst Laterality	Age at Surgery (years)	Follow-up (years)	Preop. Cyst Volume (mL)	Postop. Cyst Volume (mL)	Galassi Type	Postoperative Outcome
1	Male	Delayed motor development, language difficulties	Left	13	3	50	22	1	Improved
2	Male	Paresthesias, headache, dyslexia, difficulties concentrating	Left	12.75	2.5	72	40	1	Headache resolved
3	Male	Epilepsy	Right	11.5	0.8	18	12	1	Seizure free
4	Female	Nausea and vomiting	Left	7.75	1.5	78	39	2	Nausea and vomiting resolved
5	Female	Language and concentration difficulties	Left	11	2.5	21	11	1	Improved
6	Female	Language difficulties, headache	Right	6	2	341	207	3	Unchanged
7	Male	Tics and paroxysmal perceptual difficulties, headache and retroorbital pain	Left	7.75	1.5	18	7	1	Improved regarding pain. Tics unchanged
8	Male	Attention Deficit Hyperactivity Disorder and headache	Left	10.75	0.9	75	4	1	Headache resolved
9	Female	Precocious puberty and cognitive difficulties	Left	9	1	41	41	1	Unchanged
10	Male	Headache and nausea	Right	7	2	22	24	1	Headache and nausea resolved
11	Male	Epilepsy	Right	9.3	2.9	13	3	1	Seizure free
12	Male	Precocious puberty and cognitive difficulties	Right	13	2	55	17	1	Improved cognition.
13	Male	Non-verbal learning difficulties	Right	Diagnosis at 7.5 years	-	98	-	2	Declined surgery
14	Female	Attention Deficit Hyperactivity Disorder	Left	Diagnosis at 7.7 years	.	14	-	1	Normal cognitive test results No surgery
15	Female	Headache secondary to hygroma, concentration difficutlies	Left	Diagnosis at 12.7 years	4	37		1	Normal cognitive testing Acute hygroma evacuation but no elective surgery for cyst

Abbreviations. N/A: Not available. Preop: preoperative. Postop: postoperative.

**Table 2 jcm-09-00850-t002:** Pre and postsurgical neuropsychological test results.

	Presurgical mean (SD)	Postsurgical mean (SD)	Difference (95% CI)	*p*
WISC-IV				
FSIQ	84.8 (8.7)	93.0 (11.9)	8.2 (3.2–13.2)	0.005
VCI	86.7 (13.1)	94.7 (13.1)	8.0 (4.2–11.8)	0.001
PRI	99.1 (11.4)	103.8 (9.6)	4.7 (−2.6–12.0)	0.180
WMI	82.5 (9.7)	84.5 (15.7)	2.0 (−5.5–9.5)	0.566
PSI	81.5 (10.4)	92.5 (12.6)	11.0 (4.4–17.6)	0.004
NEPSY-II				
MF, IR	9.5 (3.6)	10.8 (3.7)	1.3 (0.1–2.5)	0.033
MF, DR	9.5 (3.3)	11.8 (3.2)	2.3 (0.6–4.1)	0.018
MS	9.5 (5.2)	10.4 (3.5)	0.9 (-4.1–5.9)	0.696
MW	10.7 (2.8)	12.7 (2.9)	2 (0.3–3.7)	0.027
	**Presurgical**	**Postsurgical**		***p***
FAS				
Raw scores, median (range) Age-standardized, mean (SD)	17 (1–28) −0.7 (0.8)	17 (5–35) −0.3 (2.1)		0.168 0.524
BNT				
Raw scores, median (range) Age-standardized, mean (SD)	35 (22–44) −2.1 (2.6)	33 (31–45) −1.7 (1.4)		0.023 0.473
RCFT				
3 min	33 (< 20–52)	32 (< 20–58)		0.646
30 min	34 (< 20–51)	30 (< 20–55)		0.414

Abbreviations: SD: Standard Deviation. 95% CI: 95% Confidence Interval. WISC-IV: Wechsler Intelligence Scale for Children version IV. FSIQ: Full Scale Intelligence Quotient. VCI: Verbal Comprehension Index. PRI: Perceptual Reasoning Index. WMI: Working Memory Index. PSI: Processing Speed Index. NEPSY-II: A Developmental Neuropsychological Assessment II. MF, IR: Memory for Faces with Immediate Retention. MF, DR: Memory for Faces with Delayed Retention. MS: Memory for Story. MW: Memory for a list of Words. FAS: F-A-S test from the Neurosensory Center Comprehensive Examination for Aphasia. Age-standardization centered on 0, with 1 point = 1 SD (z-score). BNT: Boston Naming Test. Age-standardization centered on 0, with 1 point = 1 SD (z-score). RCFT: Rey Complex Figure Test; retention at 3 min and 30 min delay.

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
