# Peer review of "Cognition in Children with Arachnoid Cysts"

_jcm, 2020, doi:10.3390/jcm9030850_

Round 1
Reviewer 1 Report
Re.: Manuscript ID: jcm-749492. Ulrika Sandvik et al. Cognitive Effects of Neurosurgical Interventions for Temporal Arachnoid Cysts in Children
Thank you for allowing me to review this prospective study on cognition in children with arachnoid cysts (AC). The article is of importance, mainly because very little has been done before on this topic, whereas several aspects of cognition has been studied in adults.
I have the following specific comments in a non-prioritized way:
- I suggest that the authors use subheadings, both in the Methods, Results, and perhaps also the Discussion sections; my suggestion would be to discuss each set of cognitive tests under a subheading that gives the name of the test, in the Methods describe the test, and what information it gives; in the Results what the results of the test were and in the Discussion perhaps group the tests probing similar aspects of cognition.
- In my opinion, the title doesn’t reflect all the content of the article, as it emphasises only the effect of surgical decompression on cognition, but not the cyst’s preoperative effect on cognition – before decompression. Perhaps “Pre- and postoperative cognition in children with arachnoid cysts” would increase the interest in potential readers?
- The last paragraph on page 4 appears to cover the same information as the second paragraph on page 5.
- For image 1 a & b, I suggest indicating the patient number in Table 1.
- I assume that the tests were age adjusted, but this could probably be expressed more clearly, above all whether the authors changed test versions from pre- to postoperative testing, As 10-12 months (average 11.6 months) elapsed between the preoperative and the postoperative tests: did they change test version between the two tests or did they give the same test (age-wise)? I believe not, but the authors should clarify this.
- The language appears to me to be a little indirect some places. Instead of using the phrase “should be surgically addressed”, why not simply say whether they should be “operated” or “surgically decompressed”?
- In the 6th line of the introduction “were” should be exchanged with “where”.
- In Table 1, I suggest using only one way of indicating “language difficulties”.
- First line page 7: “performed preoperatively” sounds better to me than “preoperatively performed”.
- First line, 3. paragraph of Discussion: “several papers reporting of possible associations “. Either delete “of” or use “on”.
- Line 13 page 8:”to exclude a practice effect a recent study” – put in “of” between “effect” and “a” -” to exclude a practice effect of a recent study”?
- The rest of the same sentence is also difficult to understand: what is a “middle school sample of eight-year-old American children”?
Author Response
We would like to thank Reviewer 1 for his/her valuable opinions.
- The subheadings are now adjusted accordingly with different subheadings.
- The title of the manuscript has now been adjusted to a more clear title
- The paragraph that repeated the information regarding ethical approval has been removed.
- The patient number is indicated according to the patient number in Table 1
- We have re-written this paragraph to clarify that the same versions of the tests were used consistently and that the tests are age-adjusted
- The language has been revised to avoid indirect phrasing
- The typo is corrected
- Table 1 has been revised for more consistent naming
- The typo is corrected
- The language has been corrected
- The language has been corrected
- We agree with Reviewer 1, the paragraph has been re-written.
Reviewer 2 Report
This article discusses the significance of surgical intervention on symptomatic temporal arachnoid cysts in terms of cyst volume reduction and neuropsychological improvement. A prospective design was adopted and informed consent was adequately obtained by the patients' families.
The design is adequately and clearly explained but - although this does not interfere with the scientific soundness of the overall work- the lack of a control group (children with temporal arachnoid cysts and concordant symptoms, treated conservatively) attenuates the impact of the reported findings. As far as the case of child 15 is concerned, it is stated that the symptoms in such case were due to acute hygroma rather than cyst so the enrollment of this patient in the study may be questionable. Finally, volume measurement have been carried out by a single operator -while two independent measurements would have been more robust from a methodological point of view.
Apart from such minor inconsistencies, the study is well described and well written and its limits are adequately addressed. The results appear as relevant and are robustly integrated with the already existing -and correctly reviewed- literature.
Author Response
We would like to thank Reviewer 2 for his/her valuable opinions
We had some discussions regarding which children to include. The situation with a child having a hygroma from a ruptured arachnoid cyst and also having pre-existing cognitive difficulties reflects our clinical reality where the incidental findings need to be correlated with clinical status. Because the child had a history of cognitive problems we think she should be included. We have revised the paper regarding this.
We agree that two independent measurements would have been more robust. However, in this case the measurements themselves were conducted by a software (Iplannet, Brainlab) in order to minimize any bias.
We have thoroughly looked through the paper for any other typos and inconsistencies and would once again like to thank Review 2 for his/her time and effort.
Reviewer 3 Report
Sandvik et al report a series of patients with temporal arachnoid cysts managed both conservatively and surgically. Neurocongitive features were studied in depth with the help of a pediatric neuropsychologist. Fifteen patients were enrolled, of which 11 underwent surgery. All patients were referred in the context of concern for being symptomatic AC by other providers. The authors report improvement in FSIQ, VCI, and PSI. The authors propose that arachnoid cysts in this area may have a negative impact greater than previously assumed. My comments are the following:
- It appears that the statistical findings were in comparison to pre and post-operative patients of the same identity. It would be helpful to also report the change in the non operative patients at the time of diagnosis and most recent follow-up, even if only some of them have data.
- Since patient 15 did not have surgery for the problem in question, I would change the age at surgery to the Diagnosis at ## years option
- Was there any analysis of the relationship between cyst volume reduction and cognitive improvement besides FSIQ? That value is so close to significance that it would be worth commenting on the relationship trend seen.
- It would be of value to comment on the durability of these tests other than FSIQ relative to age, i.e. a direct statement that these tests should not change within a year interval of testing by development alone.
Author Response
We would like to thank Reviewer 3 for his/her valuable opinions
All the available data is reported in the paper. The conservatively managed children have not been followed with any new radiology or cognitive evaluations.
The data regarding Patient nr 15 has been corrected.
The significant findings regarding cyst volume and WISC-IV subscores have been added.
Our writing regarding the durability of the tests has been revised according to Reviewer 3's suggestions.